# Effects of Waterlogging at Flowering Stage on the Grain Yield and Starch Quality of Waxy Maize

**DOI:** 10.3390/plants13010108

**Published:** 2023-12-29

**Authors:** Huan Yang, Xuemei Cai, Dalei Lu

**Affiliations:** Jiangsu Key Laboratory of Crop Genetics and Physiology, Jiangsu Co-Innovation Center for Modern Production Technology of Grain Crops, Yangzhou 225009, China

**Keywords:** waterlogging, waxy maize, grain yield, starch granule size, pasting viscosity

## Abstract

Waterlogging is a common abiotic stress in global maize production. Maize flowering stage (from tasseling to silking) is more fragile to environmental stresses, and this stage frequently overlapped the plum rain season in the middle and lower reaches of Yangtze river in China and affect the yield and quality of spring-sown maize severely. In the present study, the soil moisture content under control and waterlogging conditions at the flowering stage was controlled by a negative-pressure water supply and controlling pot device in a pot trial in 2014–2015. The grain yield, starch content, and starch structural and functional properties under two soil moisture levels were compared using Suyunuo5 (SYN5) and Yunuo7 (YN7) as materials, which are the control hybrids of National waxy maize hybrid regional trials in Southern China. The results observed that the grain yield was reduced by 29.1% for SYN5 with waterlogging due to the decreased grain weight and numbers, which was significantly higher than that of YN7 (14.7%), indicated that YN7 was more tolerant to waterlogging. The grain starch content in YN7 was decreased by 9.4% when plants suffered waterlogging at the flowering stage, whereas the content in SYN5 was only decreased in 2014 and unaffected in 2015. The size of starch granules and proportion of small-molecule amylopectin with waterlogging at the flowering stage increased in SYN5 and decreased in YN7 in both years. The type of starch crystalline structure was not changed by waterlogging, whereas the relative crystallinity was reduced in SYN5 and increased in YN7. The pasting viscosities were decreased, and the pasting temperature was unaffected by waterlogging in general. The gelatinization enthalpy was unaffected by waterlogging in both hybrids in both years, whereas the retrogradation enthalpy and percentage in both hybrids were reduced by waterlogging in 2014 and unaffected in 2015. Between the two hybrids, YN7 has high pasting viscosities and low retrogradation percentage than SYN5, indicated its advantages on produce starch for more viscous and less retrograde food. In conclusion, waterlogging at the flowering stage reduced the grain yield, restricted starch accumulation, and deteriorated the pasting viscosity of waxy maize. Results provide information for utilization of waxy maize grain in food production.

## 1. Introduction

In the current climate change scenario, agriculture faces increasing instability with extreme weather events, leading to considerable yield losses [1]. Waterlogging is a prevalent environmental adversity that suppresses maize (*Zea mays*. L) productivity and degrades grain quality; it is predicted to increase in magnitude and frequency along with global warming [2]. In South and Southeast Asia alone, 18% of the total maize growing area is frequently affected by waterlogging, which dominates 25–30% of annual production losses [3]. Maize is a rainfed crop and is considered vulnerable to waterlogging when the field soil moisture content is higher than 80%, thereby affecting plant growth and development [4]. Among different growth stages, ranking of the daily water requirement of maize from high to low was the flowering to the silking stage, the tasseling to the flowering stage, the jointing to the tasseling stage, the silking to the harvesting stage, and the lowest at the sowing to the emergence stage [5]. Among different stages of maize, waterlogging occurred at the jointing stage had the greatest effect on maize growth and grain yield, followed by seedling, tasseling, and milk maturity stages, and the 100-grain weight and grains per ear had the largest effect on grain yield, and the severity gradually increased with the lengthening of waterlogging duration [6]. In comparison, another study reported that 6-day waterlogging at the seedling stage was more severe than occurred at the jointing and tasseling stages [7,8]. In the middle and lower reaches of the Yangtze river, plum rain season usually prevails in June, which overlapped with the spring-sown maize late growth stages, and severely affects plant pollination and grain filling, and ultimately results in yield loss. Waterlogging around the flowering stage induces the increase in lodging risk by decreasing the stem diameter, rind penetration strength, and transverse bending strength of the third base internode [4]. Waterlogging also decreases the chlorophyll and carbohydrate contents [9] and the nitrogen content in different organs at silking and maturity stages [10]. Waterlogging also disturbs carbon-nitrogen metabolism, breaks plant endogenous hormone balance, accelerates leaf senescence, eventually results in a significant reduction in photosynthetic capacity and maize grain yield [11,12,13]. 

Starch is the dominant storage substance (~70%) of the grains, and it determines the grain quality. Among different processing qualities, pasting and thermal properties are the two most important indexes; pasting properties reflect the starch viscosity during the heating and cooling stages; thermal properties, including gelatinization and retrogradation, which reflect the thermal stability and are determined by starch structural parameters such as granule size, amylopectin chain length, molecular weight distribution, etc. Both pasting and thermal properties can provide information for starch production and utilization [14]. Waterlogging affects grain starch formation, deposition, and structural and functional properties. Waterlogging starting at the anthesis stage decreases the amyloplast number in the wheat endosperm and advances the programmed cell death in endosperm cells [15]. Post-anthesis waterlogging suppresses the activities of ADP glucose pyrophosphorylase and soluble starch synthase in wheat grains, increases the starch granule size, and reduces the viscosity parameters [16,17]. Meanwhile, waterlogging during grain filling decreases the grain weight and affects the milling quality, although changes in protein composition may increase or maintain the gluten strength [18]. However, the short duration of waterlogging and shading decreases the size of starch granules and increases the peak viscosity (PV) [19]. In rice, the structural and pasting characteristics of starch are affected by flooding irrigation, but the influence varies among cultivars [20]. In comparison with alternate wetting and drying, constant-flooding irrigation decreases the PV, breakdown (BD), and setback (SB) viscosities, gelatinization enthalpy (Δ*H*_gel_), and temperatures; however, the effect on starch granule size and amylopectin chain length distribution has different trends [21]. In peanuts, the contents of oil, unsaturated fatty acids, and starch in grain increased, while the contents of crude protein, soluble sugar, saturated fatty acids, and amino acids (essential, non-essential, and total) decreased under waterlogging [22]. Waxy maize is a special maize type with starch is composed of nearly pure amylopectin, which endows its high viscosity, low retrograde, and high stability than normal maize starch and widely used in some industries such as thickening agent, emulgator, adhesive, suspending agent, etc. [23]. Suyunuo5 (SYN5) and Yunuo7 (YN7) are two widely planted hybrids in Southern China, and both hybrids are the controls of the National waxy maize regional trials [24]. Our previous studies reported that waterlogging during the grain filling stage has significantly affected the starch physicochemical properties [25,26]. However, limited information is available regarding the starch deposition and functional properties of waxy maize that suffered waterlogging at the flowering stage. We hypothesize that waterlogging at the flowering stage affects starch accumulation and changes the starch structure, thereby affecting the starch physicochemical properties, such as the pasting and thermal properties of waxy maize. Results could provide a fundamental basis for using stressed starch based on the utilization of waxy maize.

## 2. Results and Discussion

### 2.1. Grain Yield

The grain weight was significantly reduced by waterlogging in both hybrids in both years, with a decrease of 8.4% and 23.5% for YN7 and SYN5, respectively (Figure 1). The grain number was also reduced by waterlogging in both hybrids in both years (6.7% for YN7 and 7.5% for SYN5), and the decrease was significant for SYN5 in 2015 and for YN7 in 2014. The reduced grain weight and number induced the yield loss in both years, and the decrease was 14.7% for YN7 and 29.1% for SYN5, indicating that YN7 was more tolerant to waterlogging at the flowering stage than that of SYN5. Yield penalty due to waterlogging at the flowering stage has been widely reported in maize and is mainly related to the decreased grain number and weight. These yield penalties may relate to the decreases in post-silking dry matter accumulation, redistribution of stored photosynthate to the grain, and the conversion capacity from carbohydrate to starch in grain [27]. This observation was caused by decreased activities of ribulose bisphosphate carboxylase and phosphoenolpyruvate carboxylase, decreased the contents of zeatin riboside, indole-3-acetic acid, and gibberellic acid but increased abscisic acid in leaf, and reduced photosynthetic rate by disordered the leaf gas exchange parameters and chlorophyll fluorescence parameters [11,12,13], which resulted in the decreased antioxidative enzyme activities, accelerating leaf senescence, and ultimately leading to the decreased biomass accumulation [27]. The yield for YN7 under different treatments was higher than that for SYN5 due to it having more grain numbers and higher grain weight, which is similar to our previous study [28]. Between two years, the grain yield was lower in 2014, which may be due to this year suffering high temperatures and low sunlight during grain filling stages.

### 2.2. Starch Content

The starch content in YN7 was significantly decreased by waterlogging (6.3% and 12.5% in 2014 and 2015) in both years; the value in SYN5 was unaffected in 2015 and decreased by 6.7% in 2014 (Figure 1). The unaffected starch content in SYN5 in 2015 may be due to the severe yield decrease this year, and the shrunken sink makes the surviving grains receive a similar source. Our previous study on fresh waxy maize (harvested at the milk stage) observed that the starch content was increased due to the accelerated grain filling and shortened grain filling duration [28], as the starch content increased gradually with grain filling proceeding, but the shortened filling duration results in lower starch content in the grain that harvested at maturity. Zhou et al. [17] observed that post-anthesis waterlogging reduced the wheat grain amylose content but did not affect the amylopectin content. The reduced starch content may be caused by the weakened activities of sucrose synthase and soluble starch synthase [27] and decreased the allocation of nitrogen and carbon assimilates at pre- and post-anthesis to the grains [29]. A study on peanuts also reported that the decreased starch content under waterlogging at the flowering stage was mainly caused by the decreased activities of sucrose synthetase and sucrose phosphate synthetase [30]. 

### 2.3. Starch Granule Size

The size distributions of starch granules under different treatments presented dual peaks (Figure 2). The average starch granule size in response to waterlogging differed between the two hybrids; the size increased in SYN5 but decreased in YN7 in both years when the plants suffered from waterlogging at the flowering stage (Appendix A). A study on rice also observed that the starch granule size in response to constant flooding irrigation management differed between the two cultivars [21]. Waterlogging during heading and anthesis stages causes damage to wheat endosperm cell structure, decrease the starch granules numbers in endosperms, result in the formation of irregular starch granules and enlarge the starch granule size [17]. Another study reported that short-time waterlogging and shading decreased the starch granules size [19]. Our previous study reported that the starch granule size of the two hybrids was decreased by post-silking waterlogging [25]. In the present study, the discrepancy in the results between the two hybrids may be due to the different features of their development; the amyloplast in the endosperm in SYN5 was formed late and less but developed quickly, whereas the starch granules in YN7 formed more number in endosperm cells [31]. Therefore, amyloplast formation in endosperm cells needs further study to clarify the discrepancy in different hybrids.

### 2.4. Starch Molecular Weight Distribution

The molecular weight distribution of isoamylase debranched starch for all the samples presented dual peaks, namely, peak 1 and peak 2 (Figure 3), consistent with those of different waxy starch resources, indicating its typical waxy starch character [32]. In the GPC profiles of amylopectin, the peak 1 fraction contains short starch chains, such as A and B chains (A + B1 chains), and the peak 2 fraction consists of long B chains with high-molecular-weight molecules [33]. The peak 1/peak 2 value with waterlogging significantly increased in SYN5 but decreased in YN7 in both hybrids, consistent with the change trends of starch granule size; this finding indicated that starch with large granule size has a high proportion of small molecular size of amylopectin branch chains [32]. A study on rice observed that the amylopectin chain length in response to constant-flooding irrigation management differed between the two cultivars [21].

### 2.5. Starch X-ray Diffraction

The X-ray diffraction (XRD) of starch provided information on the long-range molecular order and was associated with ordered arrays of double helices formed by the amylopectin side chains [34]. All the samples presented refection angles at 15°, 23°, 17°, and 18°, which present typical “A” diffraction pattern (Figure 4). The relative crystallinity (RC) with waterlogging significantly decreased in SYN5 but increased in YN7 in both years. The different responses of RC to flooding irrigation were also reported in rice [20]. In the present study, the change trend of RC was contrary to the trend of starch granule size and peak 1/peak 2 ratio, indicating the starch with large granule size and high molecular weight has low RC. Our previous study also observed that waxy maize starch with a high proportion of medium-sized starch granules has high RC [35]. Hsieh et al. [32] reported that waxy starch with a small granule size and a high proportion of peak 1 has high RC.

### 2.6. Pasting Property

The grain flour PV, SB, trough (TV), and final (FV) viscosities in SYN5 were reduced by waterlogging in both years, whereas the BD was reduced and unaffected by waterlogging in 2014 and 2015 (Table 1). The pasting viscosities of YN7 were decreased by waterlogging in 2014. The PV and BD were reduced, whereas the TV, FV, and SB were unaffected in 2015. The pasting temperature (*P*_temp_) in YN7 was unaffected by waterlogging in both years, whereas the value in SYN5 was unaffected in 2014 but decreased by waterlogging in 2015. In general, the pasting viscosities were decreased, and *P*_temp_ was unaffected by waterlogging. The decreased viscosities indicated the waterlogged waxy maize starch was less broken during heating, and the cracked granule was less re-associated after cooling to room temperatures. This finding is consistent with the observation on wheat [16,17] and our previous study on waxy maize starch [25,26]. The decreased viscosity is mainly due to the decreased starch content [36]. However, a study on wheat observed that short-time (7 d) waterlogging and shading increased the PV and *P*_temp_ but decreased the TV and FV [19]. In rice, the pasting viscosity in response to waterlogging differed among the cultivars [20]. YN7 has higher PV, BD, FV, and SB than SYN5, but they have similar TV and *P*_temp_. This finding indicated that YN7 has an advantage in producing viscous foods. 

### 2.7. Thermal Property

The gelatinization and retrogradation characteristics of waxy maize flours under different water conditions are presented in Table 2. The Δ*H*_gel_ was unaffected by waterlogging in both hybrids in both years, whereas the Δ*H*_gel_ in response to post-silking waterlogging was dependent on hybrids [26]. A study on rice also observed that Δ*H*_gel_ with constant flooding irrigation was higher than that of alternate wetting and drying irrigation, but it, compared with conventional irrigation, fluctuated between cultivar and year [21]. The *T*_o_ and *T*_p_ in SYN5 were reduced by waterlogging in both years; the two parameters in YN7 were reduced and unaffected by waterlogging in 2014 and 2015, respectively. The *T*_c_ in SYN5 was unaffected and decreased by waterlogging in 2014 and 2015, respectively. The *T*_c_ in YN7 was unaffected and increased by waterlogging in 2015 and 2014, respectively. Constant-flooding irrigation reduced the transition temperature of starch in rice [21], indicating that waterlogging reduced the stability of the starch structure [23]. 

Retrogradation occurred after the gelatinized samples were stored at 4 °C for 7 days. The Δ*H*_ret_ and *%R* in both hybrids were decreased by waterlogging in 2014 and were unaffected in 2015. Our previous study observed that the *%R* was increased by waterlogging after pollination [26]. The discrepancy may be due to the plants grown in 2015 having longer rainfall duration during grain filling (211 and 445 mm in 2014 and 2015, respectively); the adequate rainfall erased the influence of waterlogging during flowering. Between the two years, the thermal characteristics of plant growth in 2015 were significantly higher than that in 2014. The two hybrids have similar Δ*H*_gel_ and *T*_o_, but SYN5 has higher *T*_p_, *T*_c_, Δ*H*_ret_, and *%R* than YN7, which endows the advantage of YN7 to produce low retrograde food.

## 3. Materials and Methods

### 3.1. Experimental Design

A pot trial was conducted on the Experimental Farm at the Agricultural College of Yangzhou University, China, in 2014–2015. Two control hybrids (SYN5 and YN7) of the National waxy maize regional trial were selected as the plant materials as those two hybrids were widely planted in Southern China. Seeds were sown on March 15 in both years. The two seedlings at the one-leaf stage were transplanted to pots (38 cm height and 43 cm diameter), and one seedling at the jointing stage was left. Each treatment included 30 pots. The soil organic matter, total N, mineral N, test P, and test K contents were 19.5 g/kg, 1.23 g/kg, 69.1 mg/kg, 35.7 mg/kg, and 86.2 mg/kg in 2014, and were 17.3 g/kg, 1.21 g/kg, 100.2 mg/kg, 33.6 mg/kg, and 124.5 mg/kg, respectively. The plants were supplied with 10 g of compound fertilizer (N:P_2_O_5_:K_2_O = 15%:15%:15%) at transplanting time and 6 g of urea (N = 46%) at the jointing stage [25]. The weeds in the pot were manually removed. The mean temperature, rainfall, and sunlight durations during plant growth in 2014 and 2015 were 21.73 and 21.00 °C, 426 and 730 mm, and 599 and 500 h, respectively (Appendix A).

Soil moisture content was controlled by a negative-pressure water supply and controlling pot device [37] by setting the water supply tension of the device at different values [24]. Before the tasseling, the soil’s relative moisture content was set at 75%. At the flowering stage (from tasseling to silking), the soil relative moisture content for control (CK) and waterlogging (WS) was set at 80% (75~85%) and over 100% (10 mm water level aboveground). Stress was terminated after the ears were manually pollinated. During treatment, the plants were covered with a transparent canopy that was 5 m high aboveground to avoid the influence of rainfall. After treatment, the soil’s relative moisture content was reset to approximately 75% until maturity.

### 3.2. Grain Yield

The grains were harvested at maturity (about 40 days after pollination), and the grain number per ear was counted. The grains were manually stripped from the cobs, and grain weight (mg) and grain yield (g/plant) were determined after sun drying.

### 3.3. Starch Content

Starch content in grains was determined with anthrone–sulfuric acid method [38]. 

### 3.4. Starch Isolation

The grains were steeped in 1 g/L NaHSO_3_ solution at room temperature for 2 days. The starches were isolated using the method described by Lu et al. [24]. The samples were rinsed with distilled water and then ground using a blender for 2.5 min. The suspensions were passed through a 100-mesh sieve. The residues on the screen were again homogenized for 1.5 min and then passed through the same sieve. The starch–protein slurry was collected in a 1000 mL wide-neck flask and allowed to stand for 4 h. The supernatant was suctioned, and the settled starch layer was collected in 50 mL centrifuge tubes and centrifuged at 3000× *g* for 10 min. The upper non-white layer was scooped. The white layer was resuspended in distilled water and stirred for 30 min before centrifugation. The isolation procedures were repeated thrice. The starch was then collected and dried in an oven at 40 °C for 48 h. The protein and ash contents in the isolated starch were determined by using the International Methods 46-10.01 and 08-17.01 of AACC. These contents were lower than 0.3% and 0.2%, indicating that the purity of starch was up to the Chinese National Standard (GB/T 8885-2017) [39].

### 3.5. Starch Granule Size

The average starch granule size (µm) was expressed in terms of the volume of equivalent spheres. The size distributions of starch granules were estimated with a laser diffraction particle size analyzer (Mastersizer2000, Malvern, Worcestershire, UK) following a procedure described in the study of Lu et al. [24]. The disperse phase was absolute ethyl alcohol. Instrument accuracy was verified by using Malvern standard glass particles. The instrument, which follows the principle of laser diffraction, can measure sizes of 0.1 and 2000 μm. 

### 3.6. Starch Molecular Weight

For isoamylase debranched starch granules, starch (5 mg) was dissolved in 5 mL of distilled deionized water in a boiling water bath for 60 min. Sodium azide solution (10 µL 2% *w*/*v*), acetate buffer (50 µL, 0.6 M, pH 4.4), and isoamylase (10 µL, 1400 U, EC 3.2.1.68, Sigma, Sigma Chemical Co., St. Louis, MO, USA) were added to the starch dispersion. The mixture was incubated in a water bath at 37 °C for 24 h. The hydroxyl groups of the debranched glucans were reduced by treatment with 0.5% (*w*/*v*) of sodium borohydride under alkaline conditions for 20 h. The preparation of about 600 µL was dried in vacuo at room temperature and allowed to dissolve in 20 µL of 1 M NaOH for 60 min. The solution was diluted with 580 µL of distilled water.

Molecular weight distribution was analyzed using a PL–GPC 220 high-temperature chromatograph (Agilent Technologies UK Limited; Shropshire, UK) with three columns (PL110–6100, 6300, and 6525) and a differential refractive index detector [40,41]. The eluent system used dimethyl sulfoxide containing 0.5 mM NaNO_3_ at a flow rate of 0.8 mL/min. The temperature of the column oven was controlled at 80 °C. 

### 3.7. X-ray Diffraction

X-ray diffraction patterns of starch were obtained with an X-ray diffractometer (D8 Advance, Bruker–AXS, Karlsruhe, Germany) operated at 200 mA and 40 kV. The scanning region of diffraction angle (2*θ*) ranged from 3° to 40° at a step size of 0.04° with a count time of 0.6 s. Relative crystallinity (RC, %) was calculated as the percentage of the sum of total crystalline peak areas to total diffractograms.

### 3.8. Pasting Property

The pasting properties of starch (1.96 g of starch added in 26.04 g of water, total weight of 28 g; 7% *db*, *w*/*w*) were estimated using a rapid viscosity analyzer (RVA, Model 3D; Newport Scientific, Warriewood, NSW, Australia) following the method of Lu et al. [26]. A sample suspension was equilibrated at 50 °C for 1 min, heated to 95 °C at 12 °C/min, maintained at 95 °C for 2.5 min, cooled to 50 °C at 12 °C/min, and maintained at 50 °C for 1 min. The paddle speed was set at 960 rpm for the first 10 s and then decreased to 160 rpm for the rest of the analysis.

### 3.9. Thermal Property

The gelatinization properties of starch were estimated by differential scanning calorimetry (DSC, Model 200 *F*3 Maia, NETZSCH, Waldkraiburg, Germany) following the method of Lu et al. [26]. Each sample (5 mg, dry weight) was loaded into an aluminum pan (25/40 microliters, D = 5 mm), and distilled water was added to achieve a starch-water suspension containing 66.7% water. Samples were hermetically sealed and allowed to stand for 24 h at 4 °C before heating in the DSC. The DSC analyzer was calibrated using an empty aluminum pan as a reference. Sample pans were heated at a rate of 10 °C/min from 20 to 100 °C. Thermal transitions of starch were defined as onset temperature (*T*_o_), peak gelatinization temperature (*T*_p_), conclusion temperature (*T*_c_), and gelatinization enthalpy (Δ*H*_gel_). Samples were stored at 4 °C for 7 days after thermal analysis for retrogradation investigations. Retrogradation enthalpy (Δ*H*_ret_) was automatically calculated, and retrogradation percentage (%*R*) was computed as %*R* = 100 × Δ*H*_ret_/Δ*H*_gel_.

### 3.10. Statistical Design

Data presented in tables and figures are the mean of three repetitions (ten plants compose a repeat). Analysis was performed using ANOVA and Duncan’s test at a significance level of *p* < 0.05 with the data processing system (version 7.05).

## 4. Conclusions

Waterlogging stress at the flowering stage decreased the grain number and weight, resulting in yield loss. The grain starch content was reduced by waterlogging. The size of starch granules was enlarged and reduced by waterlogging in SYN5 and YN7, respectively. The proportion of high-molecular weight in amylopectin and RC were decreased and increased by waterlogging in SYN5 and YN7. The grain flour pasting viscosities were reduced by waterlogging in general. The Δ*H*_gel_ was unaffected, and the transition temperatures were reduced by waterlogging in general. The Δ*H*_ret_ and *%R* in both hybrids were reduced and unaffected by waterlogging in 2014 and 2015, respectively. YN7 had higher pasting viscosity and lower *%R* than SYN5 and is superior in producing food with a more viscous taste and low retrograde. In addition, YN7 was more tolerant to waterlogging, indicating this hybrid has an advantage in growing in stressful conditions. The plants’ growth in 2015 with adequate rainfall during grain filling achieved higher grain yield, PV, BD, and *%R*. The results offer the option to choose an optimal waxy maize hybrid under normal and waterlogged conditions based on different food utilizations.

## Figures and Tables

**Figure 1 plants-13-00108-f001:**
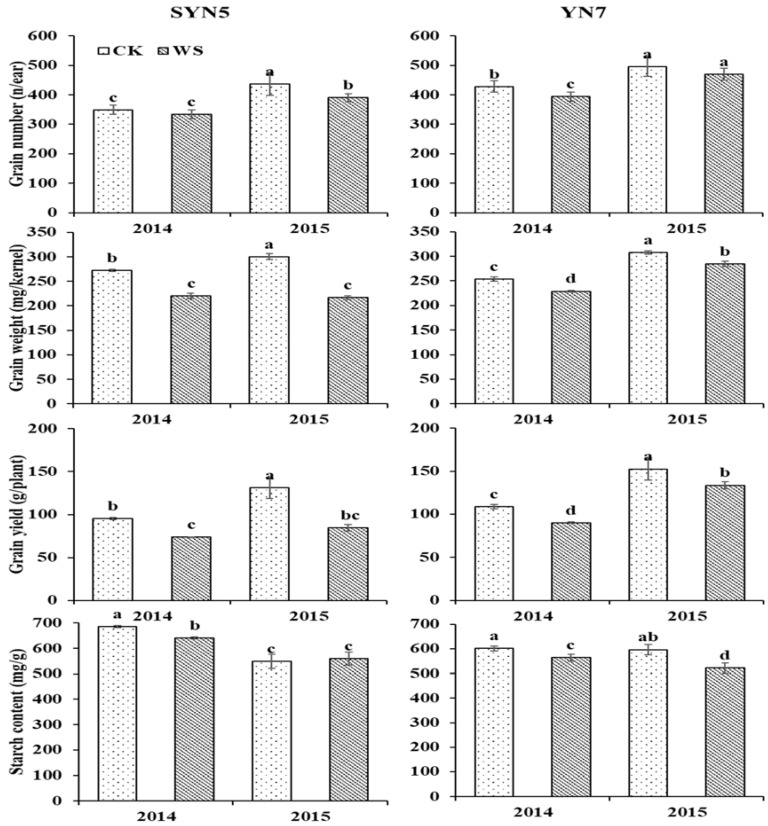
Effects of waterlogging at the flowering stage on grain yield and starch content of waxy maize. Mean value in the same column within each hybrid followed by different letters is significantly different (*p* < 0.05). SYN5, Suyunuo5; YN7, Yunuo7; CK, control; WS, waterlogging.

**Figure 2 plants-13-00108-f002:**
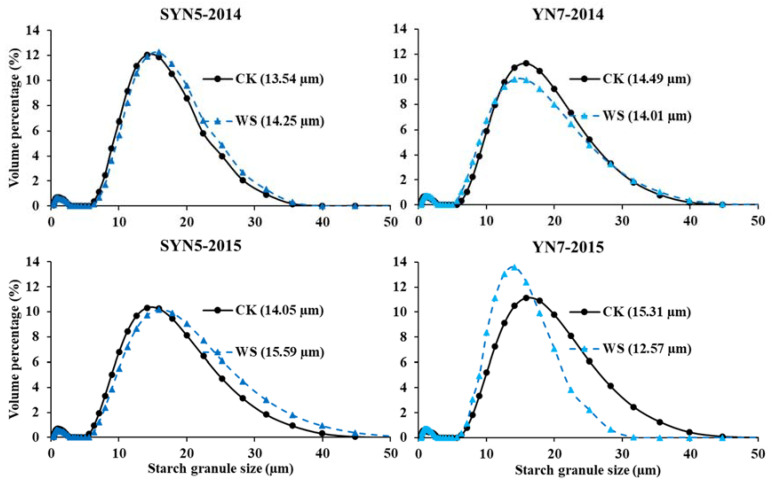
Effects of waterlogging at the flowering stage on the size distribution of waxy maize starch granules. SYN5, Suyunuo5; YN7, Yunuo7; CK, control; WS, waterlogging. The value in the bracket is the average granule size.

**Figure 3 plants-13-00108-f003:**
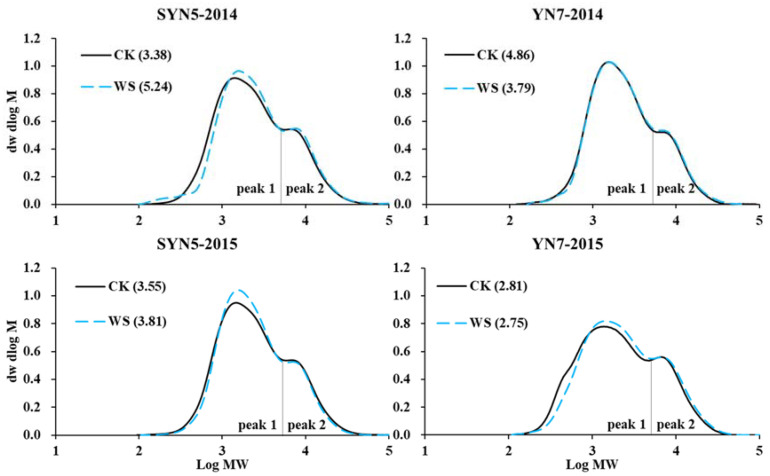
Effects of waterlogging at the flowering stage on molecular weight distribution of isoamylase debranched starch in waxy maize. SYN5, Suyunuo5; YN7, Yunuo7; CK, control; WS, waterlogging. Value in the bracket is the ratio of peak1 to peak2.

**Figure 4 plants-13-00108-f004:**
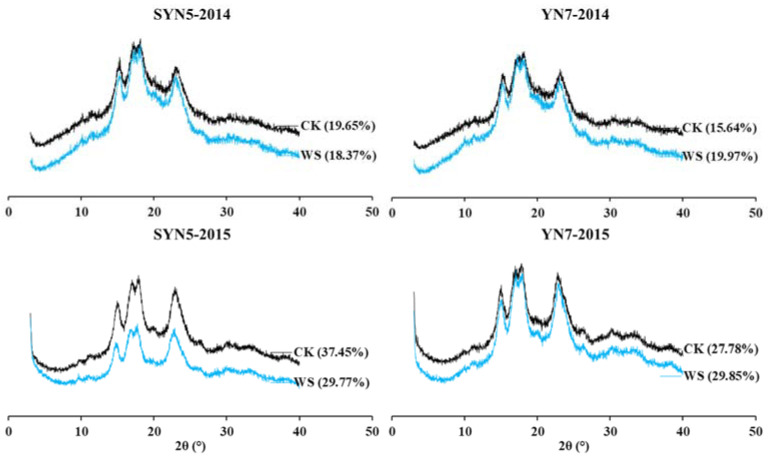
Effects of waterlogging at the flowering stage on X-ray diffraction pattern of waxy maize starch. SYN5, Suyunuo5; YN7, Yunuo7; CK, control; WS, waterlogging. Value in the bracket is the relative crystallinity.

**Table 1 plants-13-00108-t001:** Effects of waterlogging at the flowering stage on the flour pasting property of waxy maize.

Year	Hybrid	Water	PV (mPa·s)	TV (mPa·s)	BD (mPa·s)	FV (mPa·s)	SB (mPa·s)	*P*_temp_ (°C)
2014	SYN5	CK	1384 ± 1 d	1256 ± 6 a	128 ± 7 d	1660 ± 20 a	404 ± 14 a	76.1 ± 0.4 bc
		WS	998 ± 26 e	927 ± 24 e	71 ± 2 e	1181 ± 32 e	254 ± 8 de	75.3 ± 0.4 c
	YN7	CK	1378 ± 42 d	1275 ± 34 a	103 ± 8 d	1692 ± 52 a	417 ± 18 a	76.1 ± 0.4 bc
		WS	1016 ± 2 e	983 ± 3 de	33 ± 1 f	1299 ± 12 cd	316 ± 9 b	75.3 ± 0.4 c
2015	SYN5	CK	1586 ± 20 b	1134 ± 4 b	452 ± 16 c	1427 ± 6 b	293 ± 2 bc	77.9 ± 0.4 a
		WS	1477 ± 29 c	1022 ± 22 cd	455 ± 7 c	1246 ± 22 de	224 ± 0 e	76.3 ± 0.4 bc
	YN7	CK	1791 ± 4 a	1075 ± 5 bc	717 ± 9 a	1344 ± 1 bc	270 ± 6 cd	76.3 ± 0.5 bc
		WS	1659 ± 13 b	1061 ± 5 c	599 ± 9 b	1354 ± 9 bc	293 ± 4 bc	77.1 ± 0.4 ab

Mean value in the same column followed by different letters is significantly different (*p* < 0.05). SYN5, Suyunuo5; YN7, Yunuo7; CK, control; WS, waterlogging; PV, peak viscosity; TV, trough viscosity; BD, breakdown viscosity; FV, final viscosity; SB, setback viscosity; *P*_temp_, pasting temperature.

**Table 2 plants-13-00108-t002:** Effects of waterlogging at the flowering stage on the flour thermal property of waxy maize.

Year	Hybrid	Water	Δ*H*_gel_ (J/g)	*T*_o_ (°C)	*T*_p_ (°C)	*T*_c_ (°C)	Δ*H*_ret_ (J/g)	*%R* (%)
2014	SYN5	CK	8.84 ± 0.24 abc	69.6 ± 0.1 d	75.6 ± 0.0 de	82.4 ± 0.1 c	3.5 ± 0.0 a	39.7 ± 0.9 a
		WS	8.74 ± 0.22 bcd	68.5 ± 0.1 e	74.8 ± 0.0 f	81.9 ± 0.1 c	2.9 ± 0.1 b	33.0 ± 0.3 b
	YN7	CK	8.53 ± 0.12 cd	70.2 ± 0.0 cd	75.5 ± 0.0 e	82.3 ± 0.0 c	2.7 ± 0.2 b	31.9 ± 1.5 b
		WS	8.23 ± 0.07 d	68.3 ± 0.1 e	74.6 ± 0.0 f	83.1 ± 0.1 b	2.1 ± 0.2 c	24.9 ± 1.9 c
2015	SYN5	CK	8.93 ± 0.07 abc	72.5 ± 0.0 a	77.8 ± 0.2 a	84.7 ± 0.0 a	3.6 ± 0.1 a	40.8 ± 1.8 a
		WS	9.36 ± 0.11 a	70.4 ± 0.1 cd	76.2 ± 0.1 b	83.3 ± 0.2 b	4.1 ± 0.2 a	43.2 ± 2.2 a
	YN7	CK	8.83 ± 0.11 abcd	71.4 ± 0.6 b	76.0 ± 0.0 c	82.9 ± 0.1 b	3.6 ± 0.1 a	40.6 ± 1.1 a
		WS	9.16 ± 0.30 ab	70.9 ± 0.2 bc	75.8 ± 0.2 cd	83.0 ± 0.2 b	3.7 ± 0.2 a	40.0 ± 0.7 a

SYN5, Suyunuo5; YN7, Yunuo7; CK, control; WS, waterlogging; Δ*H*_gel_, gelatinization enthalpy; *T*_o_, onset temperature; *T*_p_, peak gelatinization temperature; *T*_c_, conclusion temperature; Δ*H*_ret_, retrogradation enthalpy; *%R*, retrogradation percentage. Mean value in the same column followed by different letters is significantly different (*p* < 0.05).

## Data Availability

All the data and code used in this study can be requested by email to the corresponding author Dalei Lu at dllu@yzu.edu.cn.

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
