# Peer review of "Effects of Waterlogging at Flowering Stage on the Grain Yield and Starch Quality of Waxy Maize"

_plants, 2023, doi:10.3390/plants13010108_

Round 1
Reviewer 1 Report
Comments and Suggestions for Authors
This study effectively introduces the problem (waterlogging's impact on maize production) and its relevance, providing context for the study. It specifies the research focus, which is the effect of waterlogging during the flowering stage on grain yield and starch quality of two maize hybrids, SYN5 and YN7. However, some minor issues should be addressed before publication.
The abstract could be more structured. It currently presents the findings in a somewhat scattered manner. Organizing the information with clear subsections for methods, results, and conclusions could enhance clarity. It lacks some specific details, such as the methodology used in the study, the sample size, and any statistical analysis performed. Providing these details would improve the transparency of the research. While the abstract mentions a 14.7% reduction in grain yield for YN7 and a 29.1% reduction for SYN5 due to waterlogging, it doesn't specify whether these are statistically significant differences. It's important to clarify the significance of the results.
Introduction
Line 31-34 could cite Liu et al. 2023 to support your statement. Liu, K., Harrison, M.T., Yan, H. et al. Silver lining to a climate crisis in multiple prospects for alleviating crop waterlogging under future climates. Nat Commun 14, 765 (2023). https://doi.org/10.1038/s41467-023-36129-4
Fig. 2 and Fig.3 These datasets are not fitted, so the curve cannot be used here.
Comments on the Quality of English LanguageEnglish needs to be improved by native speakers
Author Response
This study effectively introduces the problem (waterlogging's impact on maize production) and its relevance, providing context for the study. It specifies the research focus, which is the effect of waterlogging during the flowering stage on grain yield and starch quality of two maize hybrids, SYN5 and YN7. However, some minor issues should be addressed before publication.
The abstract could be more structured. It currently presents the findings in a somewhat scattered manner. Organizing the information with clear subsections for methods, results, and conclusions could enhance clarity. It lacks some specific details, such as the methodology used in the study, the sample size, and any statistical analysis performed. Providing these details would improve the transparency of the research. While the abstract mentions a 14.7% reduction in grain yield for YN7 and a 29.1% reduction for SYN5 due to waterlogging, it doesn't specify whether these are statistically significant differences. It's important to clarify the significance of the results.
Response: In the abstract: the organization was rearranged following the subsection for significance, methods, results and conclusion. The difference was also addressed. More details please refer to the MS.
Introduction
Line 31-34 could cite Liu et al. 2023 to support your statement. Liu, K., Harrison, M.T., Yan, H. et al. Silver lining to a climate crisis in multiple prospects for alleviating crop waterlogging under future climates. Nat Commun 14, 765 (2023). https://doi.org/10.1038/s41467-023-36129-4
Response: Reference was added.
Fig. 2 and Fig.3 These datasets are not fitted, so the curve cannot be used here.
Response: The presentation of the Fig. 2 and Fig.3 are the typical profiles of the starch granule size distribution and molecular weight distribution, similar to the previous published paper.
Reviewer 2 Report
Comments and Suggestions for Authors
The study "Effects of waterlogging at flowering stage on grain yield and starch quality of waxy maize" delves into crucial parameters governing the characteristics of waxy maize and how they are impacted by waterlogging. While the study holds potential interest, the manuscript's description is underdeveloped.
The manuscript outlines the challenges posed by waterlogging during the flowering stage on grain yield and starch quality. However, the primary objective and key conclusions of the study remain unclear. Several aspects need improvement before publication.
The introduction lacks depth and fails to underscore the significance of the measured parameters or the importance of understanding these characteristics for waxy maize production. It should emphasize the importance of the crop and why the measured characteristics matter. Additionally, the rationale behind selecting these specific varieties for the study should be elucidated.
The materials and methods section is too concise. It lacks descriptions of the phenological stages of different varieties, meteorological variables, and soil moisture levels throughout the study, which are essential for comprehending the results. Understanding the state of waterlogging at each stage is crucial, especially given its highlighted importance in different phenological stages.
The presentation of results should be substantiated with statistical analyses, highlighting parameters that exhibit significant differences. The discussion of results in each section is brief and does not address disparities between the studied years and varieties. Although the authors reference previous studies with the same varieties, they do not delve into why discrepancies exist or discuss the novel contributions of the current results compared to previous studies.
In the conclusions, a mere list of measured parameters is presented without clarifying the study's contribution or the importance of utilizing specific varieties and measured parameters.
While not an expert in maize cultivation, it is suggested that the manuscript should exhibit greater scientific rigor and a more in-depth discussion of the various measured parameters for coherence. This would allow the identification of critical variables and characteristics primarily affected by the treatment, facilitating exploration of potential solutions such as altering cultivation dates or utilizing less sensitive varieties.
I propose that the manuscript undergoes a major revision to address the issues identified and improve its overall coherence and impact.

Author Response
The study "Effects of waterlogging at flowering stage on grain yield and starch quality of waxy maize" delves into crucial parameters governing the characteristics of waxy maize and how they are impacted by waterlogging. While the study holds potential interest, the manuscript's description is underdeveloped.
The manuscript outlines the challenges posed by waterlogging during the flowering stage on grain yield and starch quality. However, the primary objective and key conclusions of the study remain unclear. Several aspects need improvement before publication.
The introduction lacks depth and fails to underscore the significance of the measured parameters or the importance of understanding these characteristics for waxy maize production. It should emphasize the importance of the crop and why the measured characteristics matter. Additionally, the rationale behind selecting these specific varieties for the study should be elucidated.
Response: In the second paragraph (Lines 63-69), the significance of the measured parameters was addressed. And the importance of the waxy maize was emphasized (Lines 87-90). Additionally, the rationale selecting both hybrids was elucidated (Line 90-92).
The materials and methods section is too concise. It lacks descriptions of the phenological stages of different varieties, meteorological variables, and soil moisture levels throughout the study, which are essential for comprehending the results. Understanding the state of waterlogging at each stage is crucial, especially given its highlighted importance in different phenological stages.
Response: In this section, the plant materials, meteorological variables (temperature, sunlight and rainfall) and the soil nutrients was added (Section of 2.1). And the soil moisture content was described in details (Lines 116-124).
The presentation of results should be substantiated with statistical analyses, highlighting parameters that exhibit significant differences. The discussion of results in each section is brief and does not address disparities between the studied years and varieties. Although the authors reference previous studies with the same varieties, they do not delve into why discrepancies exist or discuss the novel contributions of the current results compared to previous studies.
Response: Statistical analyses was added with supplementary table 2. The discussion of results was simply following the suggestions.
In the conclusions, a mere list of measured parameters is presented without clarifying the study's contribution or the importance of utilizing specific varieties and measured parameters.
Response: The contribution of the study, and the importance of utilizing specific varieties and measured parameters were addressed following the suggestion (Lines 355-361).
While not an expert in maize cultivation, it is suggested that the manuscript should exhibit greater scientific rigor and a more in-depth discussion of the various measured parameters for coherence. This would allow the identification of critical variables and characteristics primarily affected by the treatment, facilitating exploration of potential solutions such as altering cultivation dates or utilizing less sensitive varieties.
Response: Following the suggestion, the paper was improved as well as we can.
I propose that the manuscript undergoes a major revision to address the issues identified and improve its overall coherence and impact.
.Review:
The study "Effects of waterlogging at flowering stage on grain yield and starch quality of waxy maize" delves into crucial parameters governing the characteristics of waxy maize and how they are impacted by waterlogging. While the study holds potential interest, the manuscript's description is underdeveloped.
The manuscript outlines the challenges posed by waterlogging during the flowering stage on grain yield and starch quality. However, the primary objective and key conclusions of the study remain unclear. Several aspects need improvement before publication.
Response: The primary objective and key conclusions of the study was addressed in last sentence in the section of introduction and conclusion.
The introduction lacks depth and fails to underscore the significance of the measured parameters or the importance of understanding these characteristics for waxy maize production. It should emphasize the importance of the crop and why the measured characteristics matter. Additionally, the rationale behind selecting these specific varieties for the study should be elucidated.
Response: In the second paragraph (Lines 63-69), the significance of the measured parameters was addressed. And the importance of the waxy maize was emphasized (Lines 87-90). Additionally, the rationale selecting both hybrids was elucidated (Line 90-92).
The materials and methods section is too concise. It lacks descriptions of the phenological stages of different varieties, meteorological variables, and soil moisture levels throughout the study, which are essential for comprehending the results. Understanding the state of waterlogging at each stage is crucial, especially given its highlighted importance in different phenological stages.
Response: In this section, the plant materials, meteorological variables (temperature, sunlight and rainfall) and the soil nutrients was added (Section of 2.1). And the soil moisture content was described in details (Lines 116-124).
The presentation of results should be substantiated with statistical analyses, highlighting parameters that exhibit significant differences. The discussion of results in each section is brief and does not address disparities between the studied years and varieties. Although the authors reference previous studies with the same varieties, they do not delve into why discrepancies exist or discuss the novel contributions of the current results compared to previous studies.
Response: Statistical analyses was added with supplementary table 2. The discussion of results was simply following the suggestions.
In the conclusions, a mere list of measured parameters is presented without clarifying the study's contribution or the importance of utilizing specific varieties and measured parameters.
Response: The contribution of the study, and the importance of utilizing specific varieties and measured parameters were addressed following the suggestion (Lines 355-361).
While not an expert in maize cultivation, it is suggested that the manuscript should exhibit greater scientific rigor and a more in-depth discussion of the various measured parameters for coherence. This would allow the identification of critical variables and characteristics primarily affected by the treatment, facilitating exploration of potential solutions such as altering cultivation dates or utilizing less sensitive varieties.
Response: Following the suggestion, the paper was improved as well as we can.
I propose that the manuscript undergoes a major revision to address the issues identified and improve its overall coherence and impact.
Comments:
Introduction:
To enhance the introduction section, consider the following revisions for a more comprehensive and compelling narrative:
- Significance of Measured Parameters:
- Provide a deeper exploration of why the measured parameters are crucial for waxy maize production.
- Emphasize the practical implications of understanding these characteristics, connecting them to potential improvements in yield, starch content, and overall crop quality.
Response: A deeper exploration about the measured parameters was addressed in Lines 64-69.
- Importance of Waxy Maize and Starch Content:
- Elaborate on the importance of waxy maize in various industries, with a focus on starch content.
- Discuss the role of starch in waxy maize production, highlighting its significance in end-use applications.
Response: The special character of waxy maize is it starch was only composed by pure amylopectin, not starch content, its utilization was highlighted in Lines 87-90.
- Rationale for Variety Selection:
- Clearly articulate the rationale behind selecting the specific varieties used in the study.
- Highlight any unique characteristics or traits of these varieties that make them particularly relevant for investigating the impact of waterlogging on waxy maize production.
of understanding these temporal dynamics for accurate interpretation of the study results.
Response: the rationale of selecting the hybrids was addressed in Lines 90-92.
4.Reference to Previous Studies:
..- Explain the most significant results of previous studies that evaluated the response to waterlogging at different phenological stages of maize. Clearly state the importance of the current work in this context.
- Provide more context on how soil conditions change during these stages and the potential implications for starch physicochemical properties.
By incorporating these elements, the introduction will better convey the importance of the study, contextualize the selected varieties, and provide a more detailed understanding of the growth stages, thereby enhancing the overall clarity and impact of the research.
Response: Some references were added following the suggestions. But in the southern China, waterlogging often occurred at the late stages of spring sown maize (Lines 53-56), so we studied the effects of waterlogging at the flowering stage on maize grain yield and quality.
Material and Methods:
Material and Methods:
The methodology section requires a more comprehensive description, particularly given the study's focus on understanding how waterlogging affects the flowering stage of the crop. To enhance clarity, the following adjustments are suggested:
- Experimental Setup:
- Specify the number of pots used in the experiment, detailing any variations between years.
- Clearly state the number of plants in each pot, as well as the rationale behind this choice.
- Provide information on the number of replicates or experimental units used.
Response: The number of pots were specified, soil nutrient contents were detailed, each pot left one plants at the jointing stage, there are three replicates in the experiment, which was depicted in section 2.10.
- Phenological Stages and Soil Moisture Levels:
Provide detailed descriptions of the crop's phenological stages, emphasizing the flowering stage.
- Correspond these stages with specific soil moisture levels, indicating the range or threshold values.
Response: The flowering stage is from tasseling stage to silking stage. soil moisture we 80% and over 100% (10 mm water level aboveground). The range was provided following suggestion.
- Meteorological Conditions:
- Include a more detailed discussion of meteorological conditions in both years, such as temperature, precipitation, and humidity.
- Explain how these conditions were measured and assessed, considering their potential impact on the study outcomes.’
-Rainfall Duration Discrepancy: Elaborate on how differences in meteorological conditions, specifically the longer rainfall duration in 2015 during grain filling, were measured and assessed.
By addressing these points, the manuscript will offer a more comprehensive and transparent account of the study's methodology, allowing readers to better understand the experimental design and its implications for the results obtained.
Response: the data of meteorological conditions in both years were assessed by meteorological station, and the data was provided with a table. During treatment, the plants were covered with a transparent canopy that was 5 m high aboveground to avoid the influence of rainfall.
Review of Results:
3.1 Grain Yield:
The results lack information on the number of sampled plants or spikes studied. The impact of waterlogging on the yield of both varieties seems inconclusive regarding significant differences between them. Moreover, the reasons behind the observed differences between the two years, particularly the higher yield in 2015 compared to 2014, are not explored.
On page 2, it is mentioned that the yield for YN7 under different treatments was higher than that for SYN5 , which aligns with various studies[6] . However, this assertion appears unsubstantiated by the presented results. The absence of clear support for this claim raises questions about its validity.
Response: the number of sampled plants were addressed in material and method section. Due to the meteorological condition was different among different years, the results may be inconclusive between the two years, the reason was explained. The higher grain yield for YN7 may due to this hybrid has more grain numbers and high grain weight.
3.2 Starch Content:
The manuscript fails to specify which variety exhibits the highest starch content. The discussion on changes in starch quantity between the two varieties and across different years lacks specificity and does not align with the reported results. The significance of the differences shown in the graphs is not addressed, particularly the lower starch content in SYN5 in 2015. Additionally, if previous studies reported an increase in starch and featured the same varieties, (26) the discrepancy in results should be explained.
Response: The discrepancy was addressed following suggestions.
3.3 Starch Granule Size:
Concerns are raised about the lack of clarification on significant differences between the two cultivars. Discrepancies with previous studies regarding changes in starch granule size are mentioned, but the differences are not explained. The provided explanation for the discrepancy in results between the hybrids appears speculative and requires additional data, especially regarding conditions during different phenological stages and years.
Response: the starch granule discrepancy between the two hybrids was addressed in Lines 251-254.
3.4 Starch Molecular Weight Distribution:
The manuscript mentions pg 6 line 231 changes in peak1/peak2 values with waterlogging in SYN5 and YN7, but the significance of these differences is not addressed. The discussion lacks clarity on whether these changes are statistically significant.
Response: The statistical differences were provided in Table S2.
3.5 Starch X-ray Diffraction:
The changes in relative crystallinity (RC) with waterlogging in SYN5 and YN7 are reported, but again, the significance of these differences is not discussed. The contradiction with a previous study regarding the relationship between starch granule size and RC needs clarification.
Response: The statistical differences were provided in Table S2, and the relationship meaning was clarified in Lines 286-289.
3.6 Pasting Properties:
The presentation of results related to pasting viscosities in SYN5 is confusing, making it difficult to discern the effects on varieties or years. The discussion of different parameters indicating pasting properties is inadequate.
Response: In Lines 297-299, we clearly presented that the PV, TV, FV, and SB in SYN5 were reduced by waterlogging in both year, except that BD was only reduced in 2014. And some discussion was added following the suggests.
3.7 Thermal Properties:
Similar to the previous section, the discussion of thermal properties lacks clarity, making it challenging to understand the observed effects on the varieties and years.
Response: The discussion on the varieties and years were added in Lines 336-340.
Conclusion:
The conclusion is deemed weak and inconsistent with the description of results. The assertion that YN7 had higher pasting viscosity and low %R and is superior in producing food with viscous taste and low retrograde is not well-supported. The conclusion should better align with the specific findings and address the apparent inconsistencies in the results.
Additionally, the manuscript should highlight the most relevant results compared to previous works (e.g., experiments with similar conditions) to emphasize the unique contributions of the present study.
Response: In the agriculture experiments, the comparison of hybrids or years mainly based on statistical analysis. In the present study, YN7 had higher pasting viscosity and low %R than that of SYN5 is based on the mean value of four treatments. The conclusion was align with the results of statistical analysis.
Additionally, the MS highlight the relevant results compared to previous study works on different crops. The unique contribution of the present study is the effects of waterlogging at flowering stage on starch function. Whereas most previous works mainly focus on the waterlogging at grain filling stages on starch functional properties.
Round 2
Reviewer 2 Report
Comments and Suggestions for Authors
The manuscript enhances comprehension of the results with a more detailed discussion. Nevertheless, I find the experiment description still lacking, particularly in the omission of the number of replicas for the plots in section 2.1.
2.- The experimental setup
Response: “The number of pots were specified, soil nutrient contents were detailed, each pot left one plants at the jointing stage, there are three replicates in the experiment, which was depicted in section 2.10.”
Additionally, the lines chosen as responses do not align with the comments, which I attribute to potential issues with line numbering modification.
In my opinion, the manuscript could be published by incorporating the pertinent aspects of the experiment description.
Author Response
The manuscript enhances comprehension of the results with a more detailed discussion. Nevertheless, I find the experiment description still lacking, particularly in the omission of the number of replicas for the plots in section 2.1.
2.- The experimental setup
Response: “The number of pots were specified, soil nutrient contents were detailed, each pot left one plants at the jointing stage, there are three replicates in the experiment, which was depicted in section 2.10.”
Additionally, the lines chosen as responses do not align with the comments, which I attribute to potential issues with line numbering modification.
In my opinion, the manuscript could be published by incorporating the pertinent aspects of the experiment description.
Response: We are sorry for our blurry explains. In the trial, there are three replications for each treatment, and each repeat was composed by ten plants.
Just like the following figure.